# Dementias Platform UK Clinical Studies and Great Minds Register: protocol of a targeted brain health studies recontact database

Ivan Koychev , Simon Young, Heather Holve, Michael Ben Yehuda , John Gallacher

Psychiatry, University of Oxford, Oxford, UK

**Correspondence to**
Dr Ivan Koychev;
ivan.koychev@psych.ox.ac.uk

## ABSTRACT

**Introduction** The case for de-risking neurodegenerative research and development through highly informative experimental medicine studies early in the disease process is strong. Such studies depend on the availability of genetic as well as high-granularity, longitudinal, phenotypic data in healthy ageing individuals who can be recruited into early phase trials on the basis of their perceived dementia risk. Until now the creation of such research infrastructure has been hampered by the lack of expense and time required to gather the rich longitudinal data needed for adequate risk stratification. Dementias Platform UK (DPUK) is a public–private partnership that brings together data from over 40 cohorts in a standardised framework, which represent an until now unavailable opportunity to create such a resource through a streamlined brain health recontact platform based on existing cohorts, as well as prospectively collected data.

**Methods and analysis** The DPUK recontact platform consists of an opt-in (Great Minds, GM) and an opt-out component (Clinical Studies Register, CSR). GM requires invited DPUK cohort participants to consent to targeted recontact at the GM website and then to provide self-reported demographic and medical history information relevant to recruitment into clinical studies. Participants complete prospective browser-based and smartphone-based cognitive tests and are given the option for remote genetic and actigraphy testing. The GM data are linked to the retrospective DPUK cohort dataset, including genotypic and longitudinal phenotypic data. The CSR is a solution for cohorts explicitly allowing targeted recontact. Approved studies provide prescreening criteria on the basis of the CSR/GM dataset, and individuals meeting these criteria are offered participation directly (GM) or through the parent DPUK cohort (CSR). Descriptive statistics will be used to summarise the outcomes relevant to the number of participants engaged with the register. Its sample size is not defined but is limited by the size of the DPUK parent cohorts.

**Ethics and dissemination** The database was approved by the South Central–Oxford C Research Ethics Committee, reference 18/SC/0268 on the 27th of June 2018 and amended on the 1st of November 2019. The availability of the register to researchers will be disseminated through DPUK's official communication channels as well as national and international scientific meetings.

## Strengths and limitations of this study

► Solution for stratified recruitment into brain health studies on the basis of genetics and cross-cohort retrospective phenotypic data.
► Prospective standardised data collection: medical history, cognition, actigraphy.
► Explicit consent for recontact on the basis of perceived dementia risk.
► Recruitment on the basis of retrospective data limited by data variance between cohorts.
► Sample generalisability limited through bias towards individuals with access to and ability to interact with digital technologies (Great Minds), as well as the potential unrepresentativeness of medical research volunteers in general (Great Minds and Clinical Studies Register).

## INTRODUCTION

The case for a new generation of highly targeted and informative clinical studies in dementia is strong.[1] Although dementia is the disorder with the greatest unmet need, over the last 20 years there has been a near 100% failure in the development of new drug treatments.[2] In many cases, drugs have failed in extremely expensive phase III programmes due to lack of efficacy. While many reasons underlie this failure, they centre on an inability to demonstrate target engagement and impact in clearly defined populations, that is, an inability to recruit well characterised individuals to experimental medicine programmes and early phase trials.

The lack of success in the development of effective treatments for diseases that cause dementia is an ongoing public health challenge from which the pharmaceutical industry is disinvesting. For example, only

3.8% of experimental medicinal products in the drug discovery phase, and 1.2% in phase III target dementia, compared with 31% and 24%, respectively, for cancer.[3]

Dementias Platform UK (DPUK) is a public–private partnership funded by the Medical Research Council that aims to accelerate dementia research by bringing together longitudinal phenotypic as well as genotypic information from over 40 cohort studies.[4] Such longitudinal data are critical for the identification of participants at high risk for dementia while still in preclinical or prodromal stages.[5] Thus, DPUK is in a unique position to establish the infrastructure to offer highly characterised individuals an opportunity to consent to targeted experimental studies and clinical trials. By enabling rapid and cost-effective recruitment of accurately characterised individuals to clinical studies, a major component of dementia drug development can be substantially de-risked.

To enable a new generation of highly targeted clinical studies, DPUK has established a register of highly characterised individuals who have consented to be recontacted for dementia and brain health-focused clinical research: the DPUK Clinical Studies and Great Minds Register (CSR/GM). By combining detailed phenotyping and, where available, genomic data from cohorts, this register enables risk stratification per hypothesis at a level of detail and convenience that would not be otherwise available. The CSR includes participants with pre-existing consent for recontact through their parent DPUK cohort. Participant selection thus relies only on existing cohort data. GM is a subsample of participants in the CSR who have specifically agreed to be contacted about dementia-focused studies. In addition to relying on cohort-based data, GM collects data longitudinally (cognition through web-based and smartphone-based methods, medical conditions, level of anxiety and depression symptoms) to ensure the database critical for stratification is both up-to-date and standardised between individuals from different cohorts. Through the provision of a currently unavailable platform for stratified recruitment into experimental studies and clinical trials, GM will create increased academic opportunity and renewed industry interest in developing new dementia therapies.

## METHODS AND ANALYSIS
### Register design
The DPUK CSR/GM is a platform designed to foster highly targeted clinical studies for conditions that cause or contribute to dementia by offering participation in clinical studies to individuals who have already undergone extensive phenotyping. Conditions of interest include Alzheimer's disease but also other causes of dementia. Studies focusing on ageing processes and brain health in general will also be a priority. The CSR comprises individuals who have already provided a consent for recontact to the DPUK cohort to which they were originally recruited. DPUK therefore assists such cohorts by providing a platform for efficient

selection of participants and recruitment into studies on the basis of pre-existing cohort data. Joining the CSR will proceed on an opt-out basis, that is, participants of cohorts joining the DPUK CSR will be informed that they are joining the CSR platform unless they decline. GM, in contrast, is the part of the platform whereby participants have explicitly consented to be contacted regarding dementia and brain health studies (ie, opt-in process), and it features additional remote testing—see figure 1 for a representation of the relationship between the CSR and GM.

Prospective GM participants will be approached by their DPUK parent cohorts who will offer GM registration details (the GM webpage). DPUK cohorts may choose to offer assistance with the GM sign-up (eg, a home visit). On entering the website, the participants will have access to the information sheet and will complete a membership profile (requiring email and password). On confirming their email, participants will be given the option of signing an electronic consent (e-consent). On the website, participants will provide basic demographic and medical history information, their level of anxiety and depression symptoms, complete cognitive tasks, and will be given the option of recording their preferences for mode of contact and type of studies in which they are interested.

The GM dataset contains a limited number of variables relevant to dementia risk, including self-reported age and family history, cohort-derived genetics (APOE4 and/or polygenic risk score) and imaging data, as available. Additional variables held by the parent cohort are accessible on a study-by-study basis. To inform study exclusion criteria, the register contains information on significant comorbidity, as well as history of MRI testing. GM also gathers cognitive data recorded through the register website and a smartphone app at registration and 6 monthly intervals following consent.

### GM methods
#### Primary objective
To establish a dementia-specific clinical studies register based on existing DPUK cohort data.

#### Primary outcome
Number of DPUK cohort participants providing consent for clinical studies recontact.

#### Secondary objectives
1. To establish the sustainability of the DPUK GM.
2. To establish the capacity of clinical studies recruitment through the DPUK GM participant identification and recruitment.

#### Secondary outcomes
1. Participant with up-to-date GM data.
2. Numbers of DPUK GM participants screened and enrolled into dementia clinical studies.

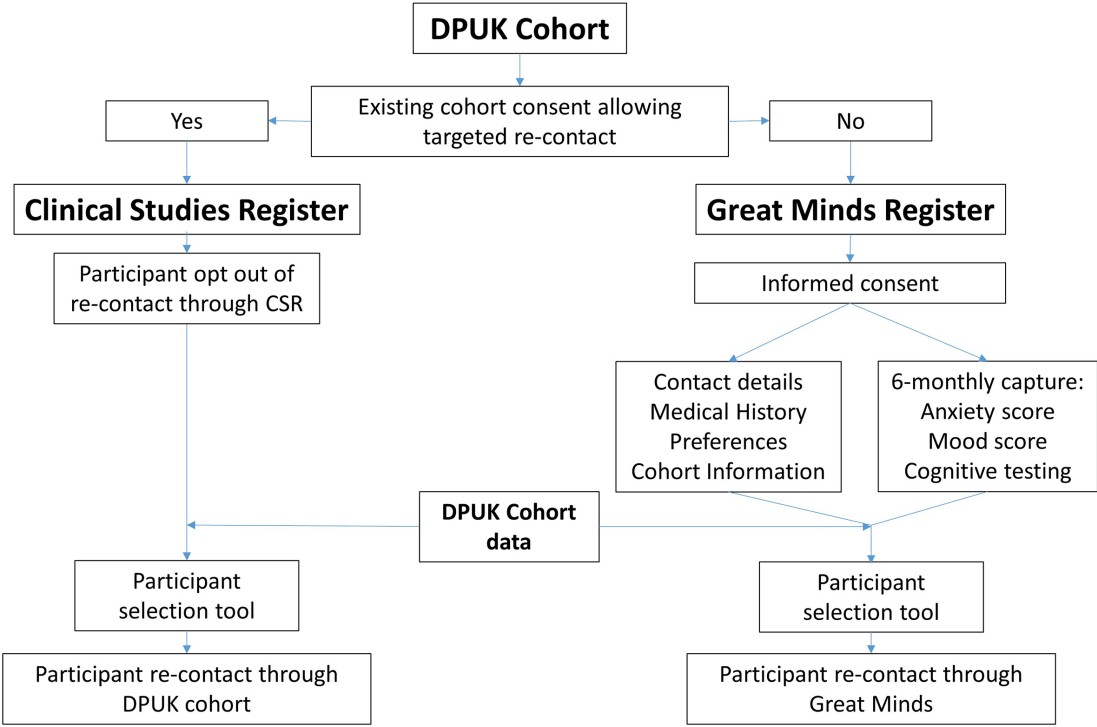

**Figure 1** A flow chart of the relationship between Dementias Platform UK (DPUK) Great Minds (GM) and Clinical Studies Register (CSR). If the DPUK cohort has an existing consent for targeted recontact, participants can get offered recontact through the CSR whereby they are also given the option to opt-out of this. The recruitment from the CSR rests solely on the existing DPUK data. GM participation is offered to participants where the cohort consent is not appropriate for DPUK recontact. Recontact through GM occurs on the basis of both existing cohort data as well as prospectively collected GM data.

## GM participants

GM comprises participants recruited to one of DPUK's constituent cohorts who, on approach by their parent cohort, agree to be included in the register.

## GM inclusion criteria

- ► Participant is willing and able to give informed consent for participation in the study.
- ► Male or female adults aged more than 18 years.

## GM exclusion criteria

- ► Participants unwilling or unable to comply with the GM register procedures.
- ► Participants who, through participation in GM Portfolio Studies, have been found to lack capacity to consent to research studies.

## Recruitment to GM

Parent cohorts (defined as being any of DPUK's constituent cohorts; see https://portal.dementiasplatform.uk/CohortMatrix for details on current DPUK cohorts) invite their participants to consider joining GM, providing information materials and DPUK contact details, as well as a unique code that is needed for registration to GM and linkage to the DPUK parent dataset. Contact details comprise a link to a GM dedicated website and a helpline. Helpline protocols include escalation procedures so that specialist questions can be addressed. On entering the website, the participant is presented with an online version of the GM information sheet. Subjects are then invited to create a profile (requiring an email, password and the unique code).

On completion of the e-consent form, the participant's personal contact details (name, address, telephone number and email address) are requested. Participants report demographic information (age, level of education), whether they currently have a cognitive disorder (dementia, mild cognitive impairment, subjective cognitive impairment), significant comorbidities (eg, diabetes, stroke, epilepsy, depression), significant MRI risks (metal in body including pacemaker), and if they have a first-degree relative with dementia. Participants are also given the option to express a preference for mode of contact, as well as the types of studies (questionnaires, observational studies, clinical trials) and interventions (MRI, Lumbar puncture (LP), cognitive or genetic testing) that they would be willing to undertake. Participants also provide information on their likely availability for study visits and preferred time of day for contact. Membership of GM is deemed to have occurred once the GM e-consent form and personal contact details have been provided. Once registered, participants are redirected to the members' area where they can access educational and GM community materials, as well as choose to withdraw from GM without needing to give a reason.

## Informed consent for GM participation

On entry to the GM website, the participant is presented with an e-consent and must personally agree to it by

typing their name in before any GM-specific procedures are performed. This requests consent for (1) recontact for recruitment to clinical studies that have received independent research ethics committee approval and have been adopted as a DPUK portfolio study; (2) recognition and acceptance that participants may be approached by studies on the basis of their dementia risk; and (3) sharing contact details with the parent cohort in order to access to existing cohort data for the sole purpose of risk stratification in relation to study recruitment.

The participant is allowed as much time as needed to consider the information, and the opportunity to discuss joining GM with the GM team through its helpline before deciding whether to participate in the register.

### GM assessments

We record cognitive information through web-based cognitive tests, as well as a cognitive smartphone-based application (if participants have an appropriate smartphone). Participants are offered optional genotyping and actigraphy assessment.

#### GM cognitive testing

For the purposes of website-based cognitive testing, we use a browser-optimised version of the Paired Associates Learning (PAL) task that is part of Cambridge Neuropsychological Test Automated Battery (Cambridge Cognition https://www.cambridgecognition.com/). The PAL task is a well-validated measure of episodic memory with a large normative database available. In the task boxes are displayed on the screen and are 'opened' in a randomised order. One or more of them contain a pattern. The patterns are then displayed in the middle of the screen, one at a time and the participant must select the box in which the pattern was originally located. If the participant makes an error, the boxes are opened in sequence again to remind the participant of the locations of the patterns. Participants complete the task at GM registration and then at 6-month intervals (they are invited to log-in again to the GM website and complete the task). The task is hosted on the Cambridge Cognition servers and selected age-adjusted and sex-adjusted outcome measures are transferred to GM on the test completion. For the purposes of participant selection, two measures of number of errors of commission are used: (1) PALTEA8, number of errors of commission when number of shapes equalled 8 plus an adjustment for the estimated number of errors on any other 8 pattern problems, attempts and recalls they did not reach; (2) PALTEA28, the total number of errors made across difficulty levels plus an adjustment for the estimated number of errors on any other 8 pattern problems, attempts and recalls they did not reach. Only demographic data (age, sex and education level), and no personal information, are made available to Cambridge Cognition for the purposes of comparing individuals' performance against normative databases.

### GM questionnaire measures

Following completion of the cognitive testing, participants are required to complete two self-reported questionnaires measuring anxiety and depression (Patient Health Questionnaire, PHQ-8[6] and Generalised Anxiety Disorder Scale, GAD-7,[7] respectively). PHQ-8 is also a brief self-reported questionnaire where participants score eight depressive symptoms 0–3 depending on their severity. GAD-7 is a 7-item scale for assessment of anxiety where items are scored 0–3 depending on the severity of each symptom.

See figure 2 for a flowchart of participant contact.

### GM smartphone-based assessments

Participants who have access to smartphones are also offered to take part in smartphone-based cognitive testing. The Mezurio smartphone application is a dementia-targeted collection of cognitive tasks (developed by a collaboration between the Oxford University Department of Psychiatry, Roche Pharmaceuticals and Eli-Lilly pharmaceuticals), released under an open-source, open-science approach.[8 9] Mezurio features a variety of cognitive tasks, including: (1) an episodic memory task, reliant on the regions of the medial temporal lobe. In this task participants are presented with photos of objects, which they are prompted to associate with one of three directions (left, right or up). Participants are asked to recollect the associations at regular intraday and interday intervals; (2) a language task in which participants are presented with consecutive frames from archive comic strips. Participants are asked to use the comic frames to narrate a story aloud; they are then asked to retell the story in immediate and delayed (~24 hours) free recall tasks. Speech data are collected via the smartphone device in order to analyse the physical, semantic and linguistic properties of the stories; (3) an executive function task requiring participants to manually tilt the phone along two axes in order to move a central counter and 'burst' presented shapes according to the designated sequence; the task is designed to maximise user engagement.

On entry to GM, all participants who choose to download the Mezurio application complete 6 days of practice, allowing them to set schedule preferences and become familiar with the presented tasks. Then, participants are offered to complete 14 further days of testing at baseline, and at 6-month intervals thereafter while in GM. During periods of testing, participants are asked to complete daily Mezurio cognitive tests (on average <8 min per day), with tasks scheduled to occur between once and three times a day.

### GM genetic testing

GM participants are offered to be genotyped remotely. Those that agree are sent a saliva testing kit which is a saliva collection device that stabilises DNA long term in a 2 or 1 mL solution. The collector is prefilled and has a simple screw-on funnel for simple saliva delivery directly into the non-toxic stabilisation buffer. The saliva collection

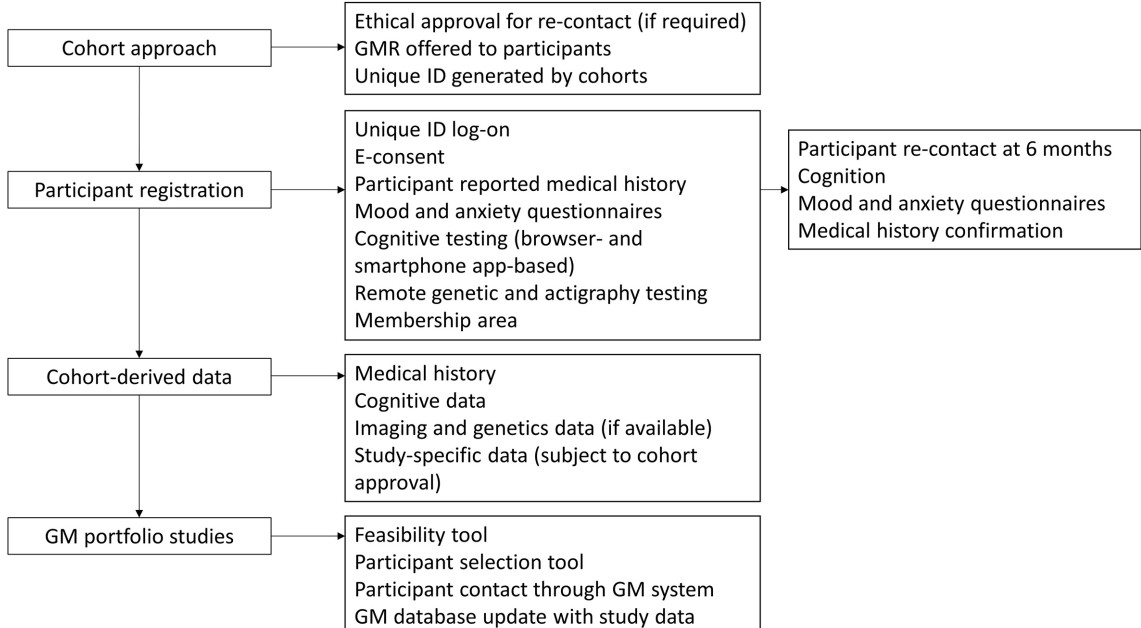

**Figure 2** A representation of the participant flow through the Great Minds (GM) platform. Participants are approached by their parent Dementias Platform UK cohort with details of GM and a unique code (ID) to register. Consenting participants provide prospective data relevant to study inclusion and these data are merged with retrospective parent cohort data to generate a dataset which is used for stratified recruitment into GM portfolio system. E-consent, electronic consent.

tube clearly indicates the volume of saliva required after which the funnel is removed, and the cap replaced. The device is fully specified for sample transport and storage. Once the sample is collected, participants are required to send it back to the study team in a prepaid envelope that is appropriate for transfer of biological samples. The samples are collected at the University of Oxford and transferred to a facility specialised in DNA extraction and analysis. The facility uploads the resulting data on the dedicated secure DPUK genetics platform. Once analysed for the purposes of GM, the samples are destroyed.

### GM actigraphy testing
GM participants are offered to take part in regular monitoring of activity through a wrist-worn actigraph. Those that agree are sent actigraphs through the post and are required to wear them for 7 days at a time. Once completed the participants are required to send back the devices back to the study team in a prepaid envelope.

### GM variables
The standardised tests included in GM will generate a standardised prospective dataset that will enrich the risk stratification options offered by the retrospective parent DPUK cohort data. For the web browser-based cognitive testing, outcome measures include the errors made by the participant, the number of trials required to locate the pattern(s) correctly, memory scores and stages completed. For the questionnaires, the outcome measures are the individual item and total item scores. For the smartphone-based cognitive testing, summary statistics are generated, including error rate for each of the tests. The genetic testing yields APOE4 carriership

status, as well as polygenic risk scores related to dementia. The actigraphs generate variables reflecting levels of activity across the day.

### GM disclosure risk procedures
The purpose of GM is to offer targeted dementia and brain health studies to participants with a wealth of phenotypic and genotypic data. Being approached by a study on the basis of variables relevant to dementia may implicitly disclose risk for dementia where this is not desired or appropriate. We seek to minimise this risk through the following safeguards: (1) joining the register is contingent on participants providing informed consent that being on the register means they may be contacted by studies on the basis of their quantified dementia risk; (2) every GM Portfolio Study has its own consent that includes safeguards on dementia risk disclosure as appropriate.

## CSR METHODS
### Primary objective
To establish a recontact platform for DPUK cohorts with consent for recontact for research.

### Primary outcome
Numbers of CSR participants screened and enrolled into clinical studies.

### CSR participants
The CSR pool of participants consists of the participants recruited to one of DPUK's constituent cohorts who have a valid consent for re-contact and who have not declined to be included in the CSR.

### CSR inclusion criteria

► Male or female adults aged more than 18 years.
► Existing valid consent for recontact for research studies through their DPUK constituent cohort.

### CSR exclusion criterion

► Explicit objection to being included in the CSR after being informed about this opportunity by their DPUK constituent cohort.

### Recruitment to the CSR

DPUK cohorts whose procedures already feature explicit consent for research studies recontact are given the option to join the CSR. DPUK cohorts that agree to take part communicate to their participants in written form notifying them that in the future they may be contacted for studies through the CSR, as well as the parent DPUK cohort. The contact materials feature information on DPUK and the CSR to inform participants' decision whether to opt-out of being contacted through the CSR. Participants are provided with practical means to decline joining the CSR while remaining a participant in the DPUK cohort. Those that do not opt-out are deemed recruited to the CSR. Membership of the CSR does not involve the collection of information in addition to the approved procedures of the DPUK cohort. Until a person is recruited into a study or joins GM, CSR members' contact details are stored by the cohort owner only (unless the cohort owners decide to share those details with the CSR for efficiency purposes).

### Informed consent for CSR participation

Only individuals with existing DPUK parent cohort consent that allows recontact for future studies are recruited to the CSR. Monitoring capacity to consent to the DPUK cohort participation and hence the CSR is subject to procedures set by the cohort itself and the relevant legislation (Mental Capacity Act 2005 in the UK). Any participants who are deemed to have lost capacity to consent to the DPUK parent cohort would not be eligible for recontact through CSR. The loss of capacity of a CSR participant will be communicated by the DPUK parent cohort to the CSR as soon as this is established and on this the participant in question will be removed from the CSR database.

### CSR assessments

Participation in the CSR does not involve any additional assessments beyond the usual procedures of the DPUK parent cohort (see DPUK Cohort Matrix for a summary of data collected for each cohort, https://portal.dementiasplatform.uk/CohortMatrix).

### RECRUITMENT TO PORTFOLIO STUDIES

Any investigator wishing to recruit from the CSR/GM into an experimental medicine study or clinical study (a 'GM Portfolio Study') must follow a multistage process. CSR and GM metadata and metadata search tools (suitably managed to preclude identification) are available to registered researchers on the CSR/GM website for study feasibility assessments prior to ethics application (see figures 3 and 4 for a graphical representation of the tool and its output). The criteria for including a study in the CSR/GM Portfolio are:

1. The protocol is finalised and research ethics committee approval has been obtained.
2. Full funding has been achieved or funding contingent on CSR/GM approval has been agreed.
3. The study falls within the remit of the CSR/GM (any medical study for the CSR; dementia or brain health-focused experimental medicine for GM).
4. They fall within the scope of the CSR/GM. For example, studies that do not require access to highly characterised individuals to answer their research question will be considered out of scope.

CSR/GM subjects eligible for a Portfolio Study are contacted by the CSR/GM and given study details. The contact details of those members who agree to receive further information on the study are forwarded to the Portfolio Study team.

### PATIENT AND PUBLIC INVOLVEMENT

We are planning to organise regular events where CSR/GM participants will be asked to provide their opinion and suggestions for improvement on (1) the processes involved in registration and continued engagement with the programme; (2) the mode of recontact; (3) the types of studies that GM should prioritise; and (4) ethical and any other issues arising from the CSR/GM involvement. We will integrate the feedback received in future changes to the programme procedures and priorities.

The design of GM and its recruitment website has been informed by a patient and public involvement (PPI) event organised by Dementias Platform UK with Alzheimer's Society members. Five people attended, represented a range of dementia experiences—caring for those with dementia, familial gene mutation and dementia diagnosis. The attendees shared their views on the value of registering to be volunteers for clinical trials, and on the risks and benefits of volunteering for a clinical trial.

The main recommendations from this PPI event were on providing clear information about data security and whether general practitioners might be contacted about outcomes. Following the event, four members signed up to join the GM PPI group which went on to review the GM website before it was launched. The recommendations from the event and the website review group have guided the creation of the register and the website, with imagery representing a more diverse community and increased signposting to data security and data privacy.

### DISCONTINUATION/WITHDRAWAL OF PARTICIPANTS FROM REGISTER

CSR/GM participants are free to withdraw from the register at any time, for any reason, without prejudice

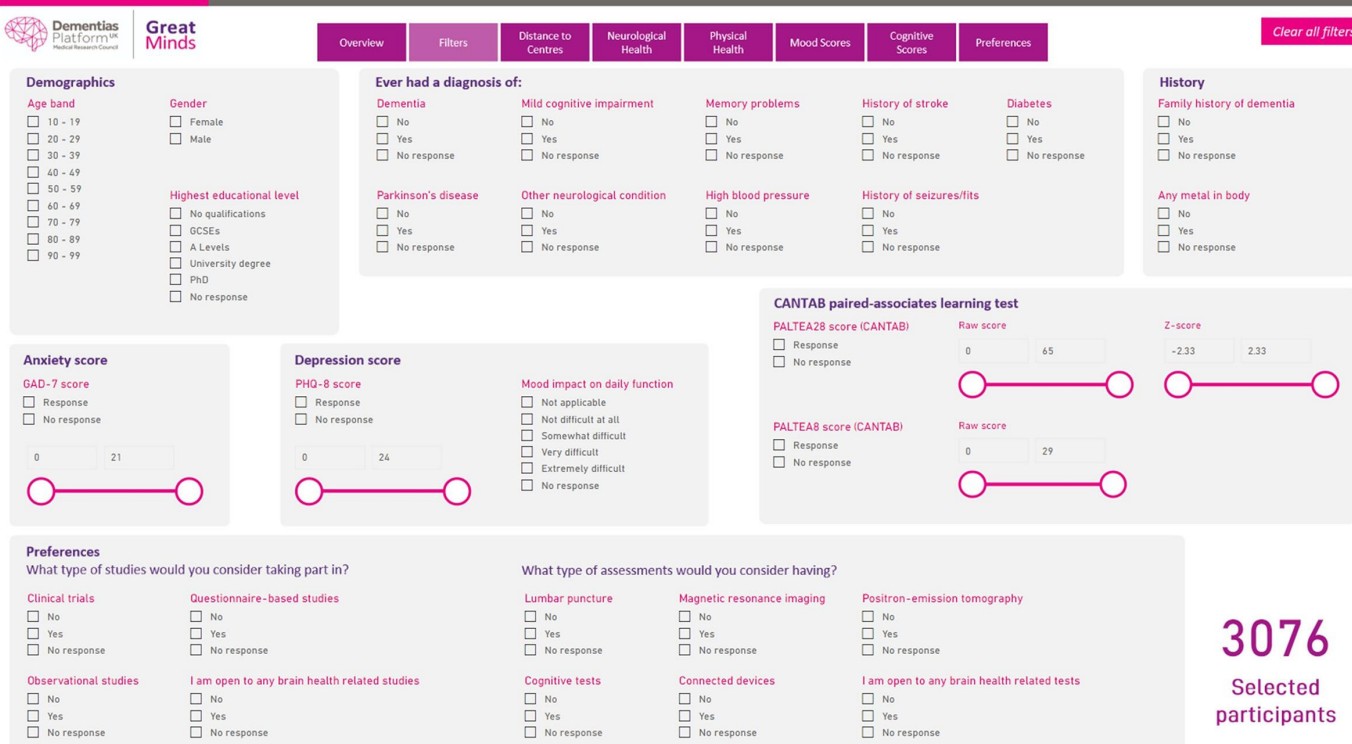

**Figure 3** A screenshot of the Great Minds feasibility tool allowing selection of participants on the basis of demographics, self-reported medical history, family history of dementia, mood and anxiety scores, cognitive test results as well as declared preferences for types of studies of interest to the participant. CANTAB, Cambridge Neuropsychological Test Automated Battery; GAD-7, Generalised Anxiety Disorder Scale; PALTEA8/28, Paired Associates Learning scores; PHQ-8, Patient Health Questionnaire.

to future care, and with no obligation to give the reason for withdrawal. GM participants can request withdrawal from the register through the GM website, emailing or phoning the GM team. CSR withdrawal occurs through the participants opting out of recontact through CSR by notifying their DPUK parent cohort. On leaving the

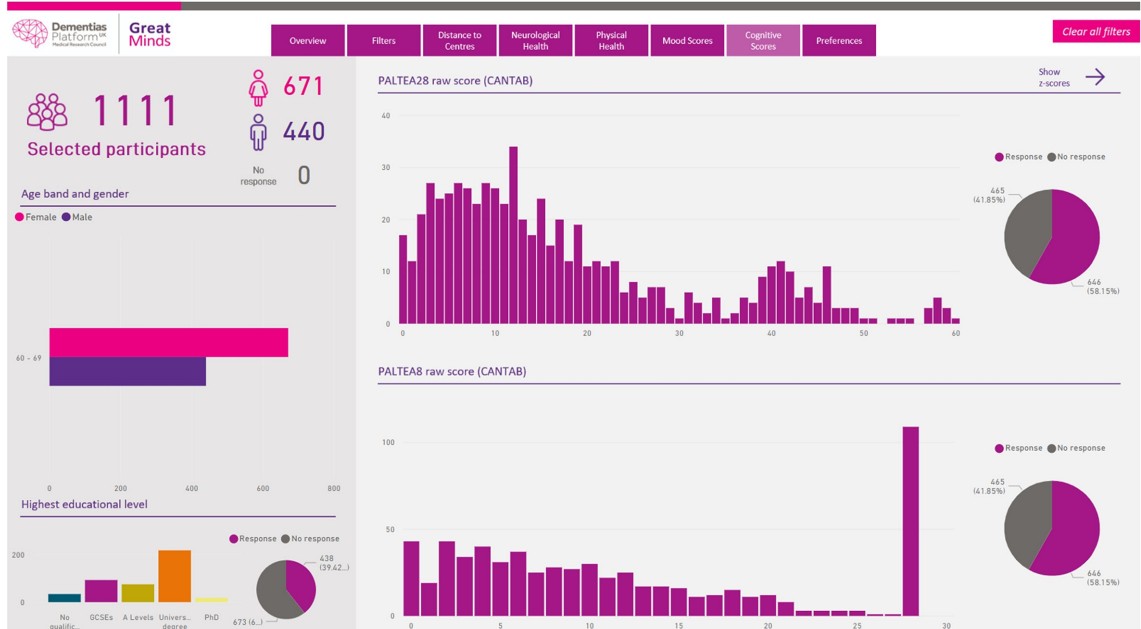

**Figure 4** A screenshot of the Great Minds feasibility tool example output (based on age stratifier) demonstrating the breakdown according to gender, educational level and cognitive performance according to Paired Associates Learning scores (PALTEA28 and PALTEA8). CANTAB, Cambridge Neuropsychological Test Automated Battery.

CSR/GM, participants stop receiving offers for participation in future studies. Loss of capacity to consent is monitored through participation in CSR/GM Portfolio Studies—sharing with us instances where a GM participant did not have capacity to provide informed consent was a condition set out in data sharing agreements with Portfolio Studies. Such participants are withdrawn from GM and are not informed of future studies.

On leaving the register, all personal identifiable data, apart from an administrative record of past membership are deleted from the register servers (ie, the unique identifier supplied by the parent cohort). In the case of GM, participants can request that data held on the servers of any suppliers or partners to GM, including Cambridge Cognition and the Mezurio app servers, are deleted as well. Participants can also request that all de-identified research data collected by GM are deleted as well.

Participants are also withdrawn from GM if they lose capacity to consent to the register. Portfolio Studies have the obligation to feedback to GM where a participant was found to lack capacity. GM staff must inform the participant of the decision to be withdrawn from GM.

## DATA ANALYSIS AND SAMPLE SIZE
### Description of statistical methods
We will use descriptive statistics to assess the primary and secondary outcome measures (number of participants registered, number withdrawing formally from GM, website and smartphone app data completion metrics, number recruited into clinical studies).

A participant selection tool has been developed to query the CSR and GM databases on an individual level according to prespecified study criteria. Identified individuals will then be offered participation in the study according to the methods described earlier.

### The number of participants
The number of participants is not defined but is limited by the total number of DPUK members. Currently DPUK facilitates access to 3 370 929 individuals from 42 cohorts.[4]

## ETHICS AND DISSEMINATION
The database was approved by the South Central–Oxford C Research Ethics Committee, reference 18/SC/0268 on the 27th of June 2018 and amended on the 1st of November 2019. The availability of the register to researchers will be disseminated through DPUK's official communication channels as well as national and international scientific meetings.

## CURRENT RECRUITMENT
Recruitment to GM began in January 2018 through the Airwave[10] and HealthWise Wales[11] cohorts. Membership as of 11th of May 2020 stands at 3055 with 1650 having completed the cognitive battery. A further 622 individuals

have downloaded and interacted with the Mezurio cognitive app. Forty-one individuals have withdrawn and requested that their data are deleted. Membership of the CSR stands at 26 436 individuals through the CSR providing a recruitment solution to the Airwave cohort.

## STRENGTHS AND LIMITATIONS
The DPUK CSR/GM platform addresses a critical unmet for preclinical dementia research by allowing stratified recruitment into brain health studies on the basis of genetics and cross-cohort retrospective phenotypic data. The dataset is also boosted by remote genetic testing and prospective standardised data collection: medical history, cognition, actigraphy. It features explicit consent for recontact on the basis of quantified dementia risk. The register is limited in its ability to recruit symptomatic individuals due to participants needing to be able to interact with digital technology to sign up and respond to study invites. Also, recruitment on the basis of retrospective data can be limited by the variability of procedures between cohorts. Finally, the GM platform is limited in its generalisability due to the reliance of participation on access and ability to interact with digital technology. While there is no such requirement for CSR, its members are nonetheless exclusively volunteers for medical research studies and therefore less likely to be representative of ethnic minorities.[12]

## CONCLUSIONS
Stratified recruitment into early phase experimental medicine studies is key to de-risking and increasing investment in neuroscience research and development. The DPUK recontact platform described provides a novel opportunity to accelerate research into novel dementia treatment through the linkage of highly characterised individuals with researchers.

**Contributors** IK and JG designed and drafted the first protocol of the register; SY and HH led on the project management of the register; MBY tested the register information technology infrastructure and provided feedback on the design. All coauthors have revised and contributed to the manuscript.

**Funding** This work was supported by the Medical Research Council Dementias grant number MR/L023784/2.

**Competing interests** None declared.

**Patient and public involvement** Patients and/or the public were involved in the design, or conduct, or reporting, or dissemination plans of this research. Refer to the Methods section for further details.

**Patient consent for publication** Not required.

**Provenance and peer review** Not commissioned; externally peer reviewed.

**ORCID iDs**
Ivan Koychev http://orcid.org/0000-0001-6813-8493
Michael Ben Yehuda http://orcid.org/0000-0002-6405-5022

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
