## [Reviewer comments · BMJ Open]

ARTICLE DETAILS

TITLE (PROVISIONAL)	Dementias Platform UK Clinical Studies and Great Minds register: Protocol of a targeted brain health studies re-contact database
AUTHORS	Koychev, Ivan; Young, Simon; Holve, Heather; Ben Yehuda, Michael; Gallacher, John

VERSION 1 – REVIEW

REVIEWER	Shireen Sindi Karolinska Institute. Sweden
REVIEW RETURNED	29-Jun-2020

GENERAL COMMENTS	This is a well written protocol article, and the Dementias Platform UK Clinical Studies and Great Minds registers have the potential to offer unique and valuable resources for stratified recruitment into clinical studies, and better selection of target populations. This is especially important as recent dementia prevention interventions have highlighted the importance of carefully selecting the target samples based on demographic, lifestyle, genetic and vascular risk factors. The following are minor comments and edits: METHODS: - The section starting with 'Primary Objective (GM)', can benefit from some re-structuring and modifications of subheadings. It is sometimes difficult to follow when the description goes back and forth between registers. The following order of items may be easier to follow: I) Great Minds (GM) Register: - Primary ObjectiveSecondary Objective-Primary OutcomeSecondary Outcome-ParticipantsInclusion CriteriaExclusion CriteriaRecruitment-Assessments
---

	Cognitive testing Questionnaire measures Smartphone based assessments Genetic testing Actigraphy testing -Variables -Disclosure risk procedures. II) Clinical Studies (CS) Register -Participants Inclusion Criteria Exclusion Criteria -Recruitment -Assessments - Is there a reason for CS Register currently not having objectives and outcomes, similar to GM? - Regarding the inclusion and exclusion criteria for CSR, it states “Existing valid consent for re-contact for research studies through their DPUK constituent cohort.” It is unclear what the procedures are for example if a DPUK developed dementia since they provided the DPUK consent. Assuming the previous consent is no longer valid, how is this handled? - Under ‘Recruitment to the CSR’, the following sentence is not entirely clear “Those that agree communicate to their participants in written form notifying them...”. Who does “those” refer to? Is it not the participants? It seems there is an error in the sentence. - Under ‘Recruitment to GM’, the first sentence describes “Parent cohorts”. It may be useful to provide a list (perhaps as an appendix) with the DPUK constituent cohorts, or a website link for example. - Under ‘CSR Assessments’, it states “Participation in the CSR does not involve any additional assessments beyond the usual procedures of the DPUK parent cohort”. As some readers may not be familiar with the procedures of the DPUK parent cohort, please provide a sentence or two with an overall summary of the categories of procedures included. - Under ‘GM Questionnaire measures’, please spell out PHQ-8 and GAD-7 and provide references. - Under ‘GM Smartphones based assessments’, please ensure consistency of capitalisations for the sentences following points (i, ii, iii).
--	---

	- Under 'GM Disclosure risk procedures', it states "joining the Register is contingent on participants providing informed consent that being on the Register means they may be contacted by studies on the basis of their perceived dementia risk;". Is this based on their "perceived" risk, or is the risk measured / quantified? - The abbreviation PPI needs to be defined. - Under 'Discontinuation / Withdrawal of participants from Register', please provide more details on the procedures for withdrawal of consent. Are participants advised to call the study centre? Or is this done online? - Under the description of the statistical methods, is the abbreviation PST necessary? It is only used once in the text. - Under the number of participants, it states "The number of participants is not defined but is limited by the total number of DPUK members." Please provide the approximate total number of DPUK members. - Under 'current recruitment', would it be possible to provide the most common reasons for withdrawal of participation? - The abstract includes a section on strengths and limitations, but this is not provided in the text. Please add this section.
--	---

REVIEWER	Lori Daiello PharmD, ScM Rhode Island Hospital/Warren Alpert Medical School at Brown University
REVIEW RETURNED	07-Jul-2020

GENERAL COMMENTS	This manuscript describes the protocol for a new enhancement of the Dementia Platform UK (DPUK) project. Development of methods to quickly assemble "trial ready" cohorts to accelerate progress in dementia prevention and treatment research is a global imperative - as such, this manuscript is particularly relevant. One significant area of revision is recommended - the addition of a figure, e.g. flow diagram, to illustrate how participants enter the Great Minds (GM) cohort. There is considerable background explanation and details about the CSR cohort that seems to distract from the narrative about the GM and its assessments. A figure may help improve the readers' comprehension of the structure of the registry. A more minor point, but one of interest to those in the field, is the paragraph on page 13 that describes the disposition of participants who are not deemed competent to give consent for studies. Some additional detail here would be helpful, e.g. by what method or criteria is competency determined? If the participant is deemed incompetent to participate in a particular study, are they barred from participation in other studies or removed from the registry?
--

VERSION 1 – AUTHOR RESPONSE

Many thanks for offering us the opportunity to submit a revised version of our manuscript. We have made every effort to address the reviewers' helpful comments - please our point-by-point responses in the attached file. Also below are our responses to the questions raised by the editorial office following the submission of the revised paper:

1. Clean copy:

- Aside from the marked copy, please also provide a clean copy of your manuscript without any highlights or tracked changes and upload it as your 'main document'.

Author comments: Clean copy uploaded as requested.

2. Figure 4 missing:

- Upon checking your manuscript, I have noted that you have cited "Figure 4" but uploaded none, if there's any please upload the file on scholar one under the file designation 'Image' (except tables and please ensure that Figures are of better quality or not pix-elated when zoom in). NOTE: They can be in TIFF, JPG or PDF format and make sure that they have a resolution of at least 300 dpi. Figures in DOCUMENT, EXCEL and POWERPOINT format are not acceptable. Kindly delete the citation if there's none.

Author comments: Apologies, there are only three figures and the text has been amended to reflect this. The order of the figures was changed to reflect their appearance in the text.

3. Figure legend:

- Please include Figure legends/caption at the end of your main manuscript.

Author comments: These are now included after the references section.

4. Table 1 missing:

- Upon checking your manuscript, I have noted that you have cited "Table 1" but embedded none, if there's any please embed the Table (should be editable/in table format) in your main document file. Kindly delete the citation if there's none.

Author comments: Apologies – there was a table in an earlier version of the manuscript which was later deemed unnecessary. Reference deleted as suggested.

VERSION 2 – REVIEW

REVIEWER	Shireen Sindi Karolinska Institute, Sweden
REVIEW RETURNED	07-Sep-2020

GENERAL COMMENTS	Dementias Platform UK Clinical Studies and Great Minds register: Protocol of a targeted brain health studies re-contact database I would like to thank the authors for thoroughly addressing the comments. The manuscript reads really well. The following are the final 2 minor comments / questions I have: 1) Under '3.7 Informed Consent for CSR Participation', the authors added: 'Any participants who are deemed to have lost capacity to consent would not be eligible for re-contact through CSR.' Can the authors please specify how this loss of capacity would be determined? For example, through a phone call assessment with the participant (e.g. using phone-based cognitive screening measures)? Or through asking a study partner / family member? 2) Under 'Strengths and Limitations', please comment on the
---

	generalisability. For example, do findings from these platform generalise to non-Caucasians, non-English speakers, or those who do not have access to computers / are unable to interact with the digital technology, and may have different demographic characteristics?
REVIEWER	Lori Daiello PharmD, ScM Rhode Island Hospital/Warren Alpert Medical School at Brown University
REVIEW RETURNED	14-Sep-2020
GENERAL COMMENTS	I appreciate the consideration given to the items specified in the previous review and noted the changes in the revised manuscript. As I remarked previously, this paper provides relevant information for those engaged in cognitive aging and dementia research - particularly for those interested in using patient registries to create "trial-ready" cohorts. In my previous comments, I had suggested adding a flow diagram to describe how participants enter the Great Minds cohort. Table 1 is a schedule of assessments and provides no information on how the individual moves through the process of entering the cohort. Although this process is described in the manuscript, it is somewhat difficult to follow, partly related to the use of multiple acronyms (e.g. DPUK, CSR) and description of specified pathways to research participation through DPUK (e.g. participants who opt out of GM can remain in CSR, where they can participate in medical, but not brain health/dementia research. While not required, such a diagram would assist the reader in better understanding of the study design, including the relationships between the distinct cohorts. Thank you for the opportunity to review this excellent manuscript.

VERSION 2 – AUTHOR RESPONSE

Reviewer: 1

Reviewer Name: Shireen Sindi

Institution and Country: Karolinska Institute, Sweden

Competing interests: None declared.

Please leave your comments for the authors below

Dementias Platform UK Clinical Studies and Great Minds register: Protocol of a targeted brain health studies re-contact database

I would like to thank the authors for thoroughly addressing the comments. The manuscript reads really well.

Response: We are thankful to the reviewer for her comments.

The following are the final 2 minor comments / questions I have:

1) Under '3.7 Informed Consent for CSR Participation', the authors added: 'Any participants who are deemed to have lost capacity to consent would not be eligible for re-contact through CSR.'
Can the authors please specify how this loss of capacity would be determined? For example, through

a phone call assessment with the participant (e.g. using phone-based cognitive screening measures)? Or through asking a study partner / family member?

Response: The informed consent would be assessed through each constituent DPUK parent cohort procedures. These procedures will differ from cohort to cohort and so it is not possible to describe a specific procedure. The CSR would be informed by the DPUK parent cohort researchers of any participants who no longer have capacity to consent and at that point the participant in question will be removed from the database. We have clarified this arrangement in the text.

2) Under 'Strengths and Limitations', please comment on the generalisability. For example, do findings from these platform generalise to non-Caucasians, non-English speakers, or those who do not have access to computers / are unable to interact with the digital technology, and may have different demographic characteristics?

Response: These are important considerations and we have expanded the limitations section accordingly. Specifically, for GM we would be constrained to participants who have access to digital technology and can interact with it. This invariably biases the sample towards a sample of higher educational attainment, higher socioeconomic status. For the CSR as participation is not dependent on interaction with digital technology, this bias is less pronounced. However, the CSR still depends on individuals who have been committed to contributing to research which probably also limits the degree to which they are representative of the general population.

Reviewer: 2

Reviewer Name: Lori Daiello PharmD, ScM

Institution and Country: Rhode Island Hospital/Warren Alpert Medical School at Brown University

Competing interests: None declared

Please leave your comments for the authors below

I appreciate the consideration given to the items specified in the previous review and noted the changes in the revised manuscript. As I remarked previously, this paper provides relevant information for those engaged in cognitive aging and dementia research - particularly for those interested in using patient registries to create "trial-ready" cohorts.

Response: We are grateful for the reviewer's positive comments.

In my previous comments, I had suggested adding a flow diagram to describe how participants enter the Great Minds cohort. Table 1 is a schedule of assessments and provides no information on how the individual moves through the process of entering the cohort. Although this process is described in the manuscript, it is somewhat difficult to follow, partly related to the use of multiple acronyms (e.g. DPUK, CSR) and description of specified pathways to research participation through DPUK (e.g. participants who opt out of GM can remain in CSR, where they can participate in medical, but not brain health/dementia research. While not required, such a diagram would assist the reader in better understanding of the study design, including the relationships between the distinct cohorts.

Response: We have now included a figure describing the flow for both participant and data in GM and CSR (new Figure 1). We believe that this improves the readability of the manuscript along the helpful comments of the reviewer.

Thank you for the opportunity to review this excellent manuscript.

VERSION 3 – REVIEW

REVIEWER	Shireen Sindi Karolinska Institute
REVIEW RETURNED	07-Nov-2020

GENERAL COMMENTS	I would like to thank the authors for addressing the comments. Below are 2 minor remaining comments: Description of Figure 1: The first “on” can be removed. “The recruitment from the CSR rests on solely on the existing DPUK data” In the following sentence, should “cohort” be replaced with “participant”: “or if the cohort is interested in the prospective data collection offered by GM.”
--

REVIEWER	Lori Daiello Rhode Island Hospital/Warren Alpert Medical School at Brown University
REVIEW RETURNED	06-Nov-2020

GENERAL COMMENTS	I greatly appreciate the authors' responses. The addition of the figure to illustrate participant and data flow In GM and CSR completely addressed my previous concern. I have no further comments.
--

VERSION 3 – AUTHOR RESPONSE

Reviewer comment: Below are 2 minor remaining comments:

Description of Figure 1:

The first “on” can be removed. “The recruitment from the CSR rests on solely on the existing DPUK data”

In the following sentence, should “cohort” be replaced with “participant”: “or if the cohort is interested in the prospective data collection offered by GM.”

Author response: We have removed the first 'on' as suggested; in the following sentence the original meaning referred to the cohort owners but we agree that it was potentially confusing. We have removed the second part of the sentence as this detail is evident in the text. See final legend text below:

A flow chart of the relationship between Dementias Platform UK (DPUK) Great Minds (GM) and Clinical Studies Register (CSR). If the DPUK cohort has an existing consent for targeted re-contact participants can get offered re-contact through the CSR whereby they are also given the option to opt-out of this. The recruitment from the CSR rests solely on the existing DPUK data. GM participation is offered to participants where the cohort consent is not appropriate for DPUK re-contact. Re-contact through GM occurs on the basis of both existing cohort data as well as prospectively collected GM data.